# The Need for Universal Screening for Postnatal Depression in South Africa: Confirmation from a Sub-District in Pretoria, South Africa

**DOI:** 10.3390/ijerph17196980

**Published:** 2020-09-24

**Authors:** Kebogile Mokwena, Itumeleng Masike

**Affiliations:** Department of Public Health, Sefako Makgatho Health Sciences University, Ga-Rankuwa, Pretoria 0204, South Africa; 201811594@swave.smu.ac.za

**Keywords:** postnatal depression, Edinburg postnatal depression scale, South Africa, maternal and child health

## Abstract

Although postnatal depression (PND) is a worldwide public health problem, it is relatively higher in developing countries, including countries in Sub-Saharan Africa. Postnatal depression is not routinely screened for in primary healthcare facilities in South Africa, despite its reported compromise on mother and child health. The purpose of this study was to determine the prevalence of, as well as factors associated with, postnatal depression in a sample of clinic attendees in a sub district in Tshwane, South Africa. A quantitative and cross-sectional survey was conducted in a sample of 406 women in three healthcare facilities. The Edinburgh Postnatal Depression Scale (EPDS) was used to collect data from women who had infants between the ages of 0 and 12 months. The cut-off point for the EPDS for the depressed category was a score of 13 out of a maximum of 30. The majority of the women (57.14%, *n* = 232) had scores of 13 and above, which is indicative of postnatal depressive symptoms. On logistic regression, postnatal depressive symptoms were significantly associated with lack of support in difficult times (*p* < 0.001, 95% CI 10.57–546.51), not having the preferred sex of the baby (*p* = 0.001, 95% CI 0.37–0.58), low household income (*p* < 0.001, 95% CI 1.23–1.67), and an older baby (*p* = 0.005, 95% CI 1.21–1.49). The results show the high proportion of women who have postnatal depression but remain undiagnosed and untreated, and therefore confirm the need for routine screening for postnatal depressive symptoms in primary healthcare facilities, which are used by the majority of women in South Africa.

## 1. Introduction

Postnatal depression (PND) is a mental disorder that affects some women who have recently given birth, and negatively impacts on their ability to carry out daily routines and take care of the baby. As a public health concern, the high burden of PND has been well reported in countries with low, middle, and high income [1]. In many low- and middle-income countries, postnatal depression remains underdiagnosed and undertreated [2].

The rates of PND vary across regions and countries, with a global prevalence of up to 60% and a wide range of 3.5–63.3% in Asian countries. While some geographical areas such as Malaysia, Singapore, Denmark, Malta and Pakistan reported a low prevalence rate (0.5–9%), others including Italy, Chile, South Africa, Korea and Taiwan reported a high prevalence rate (34–57%) [3]. Although there are country variations, PND is commonly estimated to be three times greater in low- and middle-income countries than in high-income countries [4]. The prevalence of PND on the African continent also varies widely and, in a study conducted in 10 countries across Africa, Uganda reported the lowest prevalence rate of 7.1%, while the highest was 33% in Zimbabwe [5]. A recent study reported a prevalence rate of 20–34% in Zimbabwe [6], while a prevalence of 33% was reported in South Africa [7].

Among factors that profile the degree of depression are sensory processing patterns, which may be expressed as hyper- or hyposensitive. This expression is an important determinant of the appropriateness of interventions for affective disorders, including depression, which requires the consideration of the unique sensory profile of the individual, as this influences behavior and function [8,9]. In screening and health promotion, this degree of functionality is a relevant consideration for referral of the patient to the next level of care.

Factors associated with PND include the mother’s medical condition (e.g., low body mass index), psychological factors (e.g., a history of psychiatric problems and depression during the pregnancy), obstetric factors (e.g., an unplanned pregnancy), sociodemographic factors (e.g., financial problems), and cultural factors (e.g., the sex of a baby being a priority in determining the importance of the baby in the community) [8]. History of maltreatment of the mother during childhood has been identified as a risk factor to developing depression in adult life [10], which may be exacerbated during the postnatal period. Other studies have also linked maltreatment during childhood, especially emotional abuse, with other serious affective disorders, particularly suicide [11]. The triggering of these disorders during the postnatal period presents health risks for the mother and may be more severe and debilitating, and thus impact more on the ability of the mother to care for her baby. Such a child may be at risk of depression later in life, hence a repetitive cycle for the following generations.

Other factors associated with PND include lack of social support and a history of depression [9,10,11]. On the other hand, younger women between the ages of 20 and 29 years of age are reported to have a higher prevalence of PND than older women [12,13]. Other factors include mothers’ experience of stress which emanates from feeling incompetent in their parenting tasks [14,15]. Some cultural values and traditional practices in African countries are reported to exert such pressures on mothers who have recently given birth, and thus result in high prevalence of postnatal depression [15]. Additionally, immigrants, especially those who are unable to speak the language of the homeland, often report conflicting results about maternal depression, which may be explained by their understanding of “normal” [16]. 

Although PND is reported to be high in low- and middle-income countries, it is not given the attention it deserves and thus not routinely screened for in primary health clinics in Africa [17,18]. Moreover, studies conducted in Africa often focus on rural settings and not adequate attention has been given to PND in urban areas [4], thus the need for routine screening for postnatal depression in primary healthcare settings.

### 1.1. The Use of the Edinburg Postnatal Depression Scale to Screen for Postnatal Depression

The Edinburg Postnatal Depression Scale (EPNDS) is a validated scale and is probably the most frequently used scale for the screening of postnatal depression. The scale consists of 10 self-reported questions with a total score of 30, and a cut-off score of 12 implies positive PND [19]. Its appropriate use in several languages and cultures has been confirmed [20,21,22], and it is reported to include a high positive predictive value [10]. It was also previously used in a number of South African studies [23,24], and thus it is appropriate for a South African setting [17]. In the current study, EPDS was used in English and translated into Setswana and IsiZulu, which are two indigenous languages commonly used in the research setting.

### 1.2. Objectives

The objectives of the study were to determine the prevalence of, as well as factors associated with, postnatal depression in a sample of clinic attendees in a sub district in Tshwane, South Africa. 

## 2. Materials and Methods

### 2.1. Study Design

The study design was quantitative and cross-sectional, and data were collected from a sample of women who were attending postnatal care in three primary healthcare facilities in Tshwane, South Africa.

### 2.2. Study Population and Sample

The study population consisted of mothers who were eighteen years of age and above and attending postnatal care in healthcare clinics in three facilities: Laudium Community Health Centre, Bophelong clinic, and Pretoria West district hospital. Postnatal care services include growth monitoring for the babies, as well as Integrated Management of Childhood Illness (IMCI).

### 2.3. Study Setting

The study was conducted in three government funded health facilities which consisted of a clinic, a community health center, and a district hospital in Tshwane health sub-district. All three facilities offer a range of primary healthcare services, including postnatal care. The Community Health Center primarily serves clients from neighboring informal settlements.

### 2.4. Sampling and Sample Size

Simple random sampling method, in which every second woman who arrived or was already seated at postnatal clinics, was asked to participate in the study. Using the Raosoft sample size calculator for an unknown population size, a 5% margin of error, a confidence level of 95%, and a response rate of 50%, a minimum sample size of 323 was calculated.

### 2.5. Inclusion and Exclusion Criteria

Women who were 18 years old and above, were in the first year of the postnatal period, had agreed to participate in the study, and were able to provide written informed consent were included. Women younger than 18 years, whose children were older than 12 months, or who could not provide written informed consent were excluded. 

### 2.6. Data Collection

Data were collected by the researcher and/or research assistants who were trained in the ethics and use of the questionnaire. The participants were invited into the data collection venue where the details of the study were provided. The participants were allowed to ask questions and seek clarification on any aspect of the study and, if they were ready to proceed, were requested to provide written informed consent using the form provided, which was followed by the data collection questionnaire. The data collection questionnaire included sociodemographic questions of the mother, as well as questions on the baby’s health.

### 2.7. Data Analysis

The raw data from the 406 questionnaires were captured into Microsoft Excel, cleaned, and imported to STATA version 15 (Stata Corporation, College Station, TX, USA). The sociodemographic data were analyzed descriptively and expressed as mean, median, and standard deviation. The EPDS consists of 10 questions with a maximum score of 30, and a cut off of 13 was used, in which a score below 13 was categorized as not having depressive symptoms and a score of 13 and above indicated depressive symptoms. A higher score indicated more severe depressive symptoms. Logistic regression was used to explore the relationship of demographic variables with the EPDS scores, using the *p*-value of 0.05.

### 2.8. Reliability and Validity

Validity in the study was enhanced by the use of the Edinburgh Postnatal Depression Scale, which is an already valid, reliable, and standardized tool for screening of postnatal depression. The pilot study was done, and it identified logistical challenges which were modified and therefore made the data collection process smooth.

### 2.9. Ethical Considerations

Ethical clearance was obtained from Sefako Makgatho Health Sciences University Research Ethics Committee (SMUREC/H/343/2018:PG). Permission to conduct the study was obtained from the Gauteng Provincial Department of Health, The Tshwane Health District, and the management of each of the health facilities where data were collected. Written informed consent was obtained from the participants.

## 3. Results

### 3.1. Sociodemographic Characteristics of the Participants

A total of 406 respondents participated in the study. The ages of the participants ranged from 18 to 45 years, with a mean of 28 years (±SD = 5.91). The majority (61.58%) were younger than 30 years. The majority (55.06%) spoke a range of South African native languages, 32.35% other African languages, 11.36% English and 1.23% Afrikaans. The unemployment rate was high at 76.11%. The majority of the women (75.62%) had attended secondary school, while 10.43% had tertiary education. Table 1 shows the rest of the sociodemographic characteristics of the participants.

### 3.2. Characteristics of the Babies

The number of children per participant ranged from 1 to 6, with the majority (66.4%) being between 11 and 48 weeks old. Most of the babies (90.89%) were reported to be in good health, while 9.11% had some health challenges. The rest of the characteristics of the babies are reflected in Table 2.

### 3.3. The Prevalence of Postnatal Depression

The EPNDS scores ranged from 2 to 25, with a mean of 12.96 (SD = ±4.28), and, of the *n* = 406 participants, 57.14% (*n* = 232) scored ≥ 13 on the EPNDS, placing them in the category of positive PND. Figure 1 below shows the depressed and non-depressed proportions of the sample.

### 3.4. Factors Associated with Postnatal Depression

The Pearson chi-square was used to test the statistical difference between women with PND and those without (*p* value < 0.001). The majority of the sociodemographic factors were significantly associated with PND, as reflected in Table 3.

The variables that were not significantly associated with PND were the partner/husband’s threatening to hit the participant, with a *p*-value of 0.61.

Logistic regression was further run on those factors that were associated with PND on chi-square, and the results are reflected in Table 4.

The baby’s age in weeks was strongly associated with PND, with a *p*-value = 0.01 and a confidence interval of 95% at 1.23–1.67, which indicates statistical significance. The participants who had older children in weeks were more likely to report postnatal depressive symptoms than those with younger children. The participants whose babies were of their preferred sex were less likely to develop depressive symptoms, and participants who had a partner were less likely to develop depressive symptoms. Lack of support during difficult times was strongly associated with PND, with a *p*-value of 0.001 and 95% confidence interval of 10.57–546.51. Participants who do not have support during difficult times were 76 times more likely to report postnatal depressive symptoms compared to those who have support.

Household income was strongly associated with PND (*p*-value < 0.001 and 95% CI of 1.23 to 1.67, with participants whose monthly household income was less than R2000 (about $125) being more likely to develop depressive symptoms. This is aligned with the high unemployment rate in this sample, which translates into financial difficulties.

## 4. Discussion

The results of the study found a very high prevalence of 57.14% for PND among the sample, with a relatively high mean score of 12.96. This suggests not only the high proportion of women who have PND, but also the severity of this mental condition. The univariate analysis found a statistically significant association between almost all the individual variables with PND, which suggests that unfavorable social conditions explain the high prevalence of PND in this sample.

The results suggest a steady increase of PND in studies conducted in South Africa over time, with a prevalence of 45.1% reported in 2011 [25], 49.3% reported in 2014 [21], and 50.3% in 2015 [22]. As far as we know, a prevalence of 57.14% from the current study is the highest reported in any South African study. This high prevalence may be explained by the increasing difficulties in the socioeconomic status of residents, which includes high unemployment rate, low salaries, and rife domestic violence cases, which have been reported to be on the rise. The high and increasing unemployment rate in South Africa, especially among women [26], exacerbates the risk of depression which may be apparent in the postnatal period.

Variations in frequency and scale of postnatal depressive symptoms have been explained by income categories of countries [27]. African countries have been reporting higher prevalence of PND than high-income countries [28], and South Africa has reported a higher prevalence which is consistent with high income countries [29] such as some areas in Japan and the USA, whose prevalence was more than 69% [30,31]. The results of the current study are similar to previously reported findings that the majority of women who obtained high scores on the EPDS were unmarried and unemployed [22], which has a negative impact on their economic conditions. Other studies have concluded that the high income inequality in South Africa is associated with depressive symptoms, including for women in the postnatal stage [32].

The results of the current study showed a significant association between the preferred sex of the baby and PND, which was contrary to another South African study that reported no association [21]. This can be explained by cultural differences, with some cultures having a preference for male children. The current study further reported an association between the baby’s age and PND, which is similar to the findings of other studies [20,33]. However, this is contrary to even other studies which found no such association [21,25,28,34]. The development of PND is complex and the role of each factor is influenced by the existence and mix of other factors.

The study found an association between low household income and PND, which is consistent with other studies [35,36,37,38] but contrary to some [21,22], which suggests that the role of household income may be confounded by the existence and strengths of other social factors.

The study reported a significant association between PND and with whom the mothers lived, which is similar to other studies which found that the support provided by family and friends is protective against PND, compared to women who live alone or lack such support Similarly, single women were more likely to have depressive symptoms than women with partners [20,30]. In the current study, the single women who were not living with their husbands or partners were living and raising their children alone, which could have contributed to the strong association. Studies reported social isolation, living alone, and a lack of support to be associated with PND [1,4,29].

PND substantially raises the risk for adverse outcomes on all child measures [39,40,41], which include behavioral, cognitive, social, and physical growth impairment [42]. Such reports highlight the attention needed to screen for and treat mothers who present with postnatal depressive symptoms, as failing to do so exposes the affect ted children to a range of negative health and social outcomes. All the women with postnatal depressive symptoms identified in this study, as well as their babies, are at risk of negative health outcomes, which can be easily identified and interventions put in place for other women in similar positions. Screening for PND may therefore be a significant strategy to address the high child morbidity and mortality in South Africa.

## 5. Conclusions

The study identified a high proportion of mothers with postnatal depressive symptoms who need but do not have access to treatment because they have not been diagnosed. This lack of diagnosis has profound negative long-term health outcomes for both the mother and the baby. Moreover, other studies suggest that scores of 13 and above on the EPNDS may indicate borderline personality disorder or borderline personality traits [43,44], which indicates the essentiality of integrating EPNDS as a health screening tool for depression and other mental disorders. The delay in implementing routine screening for postnatal depression compromises the quality of care intended by maternal and child health services, and thus fails to support the program of improving maternal and child health outcomes. 

### 5.1. Recommendation

As with previous studies conducted in South Africa [1,21,22,25], universal screening for PND in primary health facilities is recommended, especially because the EPNDS is easily administered and can, with relative ease, be included as a standard health promotion procedure for mothers attending postnatal clinics.

### 5.2. Limitations of the Study

Because of the cross-sectional design used, the results provide a snapshot of the mental status of the women at a point in time and do not take into account factors that occurred earlier in their lives which may have contributed to depression. Examples of such factors include the maltreatment which the mother may have experienced during childhood. The design also does not enable the identification of how long the depressive symptoms were experienced. The study was conducted in a single health sub-district where the participants present with little socioeconomic variability, and the results cannot be extrapolated to other areas with a different socioeconomic profile.

## Figures and Tables

**Figure 1 ijerph-17-06980-f001:**
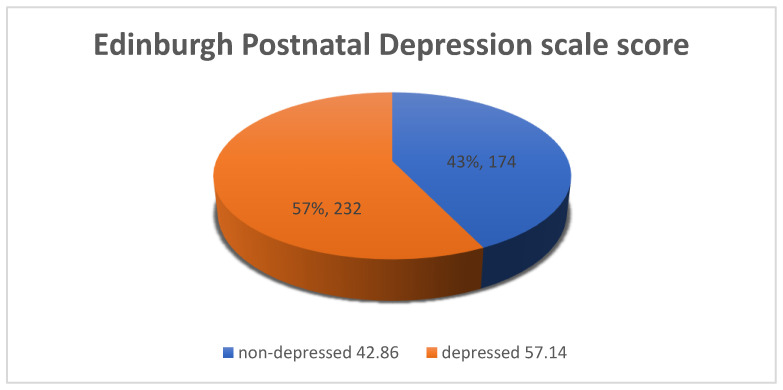
Proportion of PND (postnatal depression).

**Table 1 ijerph-17-06980-t001:** Sociodemographic characteristics of the participants.

Variable	Frequency (*n)*	Percentage (%)
**Mother’s Age**		
<30	250	61.58%
≥30	156	38.42%
	406	100%
**Home Language**		
Afrikaans	5	1.23%
English	46	11.36%
South African local languages	223	55.06%
Other African languages	132	32.35%
	406	100%
**Level of Education Completed**		
No formal education	4	0.99%
Primary	53	13.05%
Secondary	307	75.62%
Tertiary	42	10.34%
	406	100%
**Marital Status**		
Ever married	293	72.17%
Never married	113	27.83%
	406	100%
**Employment Status**		
Unemployed	309	76.11%
Employed	97	23.89%
	406	100%
**Who Does the Participant Live With?**		
Alone	37	9.11%
Partner/husband	289	71.18%
Family	79	19.46%
Friend	1	0.25%
	406	100%
**Household Income**		
R 0–499	85	20.94%
R 500–1000	114	28.08%
R 1001–2000	111	27.34%
R 2001–5000	61	15.02%
R 5000–8000	19	4.68%
R > 8000	16	3.94%
	406	100

**Table 2 ijerph-17-06980-t002:** Obstetric and baby characteristics.

Variable	Frequency (*n*)	Percentage (%)
**Parity**		
1–2 babies	297	73.15%
3–4 babies	104	25.62%
5–6 babies	5	1.23%
	406	100%
**Baby’s Age in Weeks**		
1–10 weeks	177	43.6%
11–24 weeks	124	30.54%
25–48 weeks	105	25.86%
	406	100%
**Delivery Method**		
Normal vaginal delivery	315	77.59%
Caesarean section	91	22.41%
	406	100%
**Baby Planned**		
No	126	31.03%
Yes	280	68.97%
	406	100%
**Mother Breastfeeding**		
No	76	18.72%
Yes	330	81.28%
	406	100%
**Baby’s Health**		
Good	369	90.89%
Sometimes not well	29	7.14%
Mostly not well	8	1.97%
	406	100%
**Baby’s Sex**		
Girl	206	50.74%
Boy	200	49.26%
	406	100%
**Preferred Sex**		
Girl	163	40.15
Boy	192	47.29
No preference	49	12.07
I didn’t want a baby	2	0.49%
	406	100%

**Table 3 ijerph-17-06980-t003:** Pearson chi-square comparison for participants with PND and those without.

Variable	No PND n (%)	PND n (%)	Chi2	*p*-Value
**Age of Mother**			**134.79**	**0.00**
<30 years	174 (100%)	76 (32.76%)		
≥30 years	0 (0)	156 (67.24%)		
Total	174	232		
**Home Language**			103.22	0.00
Afrikaans	5 (2.87%)	0		
English	46 (26.44%)	0		
Local South African languages	100 (57.47%)	123 (53.02%)		
Other African languages	23 (13.22%)	110 (41.51%)		
Total	174	232		
**Level of Education**			110.32	0.00
No formal education	4 (2.30%)	0 (0)		
Primary	53 (30.46%)	0 (0)		
Secondary	117 (67.24%)	190 (81.90%)		
Tertiary	0 (0)	42 (18.10%)		
Total	174	232		
**Marital Status**			117.44	0.00
Never married	0 (0)	113 (48.71%)		
Ever married	174 (100%)	119 (51.29%)		
Total	174	232		
**Employment Status**			169.9	0.00
Unemployed	77 (44.25%)	232 (100%)		
Employed	97 (55.75%)	0 (0)		
Total	174	232		
**Live With**			59.31	0.00
Alone	37 (21.26%)	0 (0)		
Family	22 (12.64%	57 (24.57%)		
Husband/partner	114 (65.52%)	175 (75.43%)		
Friend	1 (0.57%)	0 (0)		
Total	174	232		
**Occupants: Adults**			72.41	0.00
<3	141 (100%)	187 (70.57%)		
≥3	0 (0)	78 (29.43%)		
Total	174	232		
**Variable**	**No PND n (%)**	**PND n (%)**	**Chi2**	***p*-Value**
**Occupants: Children**			95.59	0.00
<3	174 (100%)	135 (58.19%)		
≥3	0 (0)	97 (41.81%)		
Total	174	232		
**Household Income**			35.22	0.00
R < 2000	158 (90.80%)	152 (65.52%)		
R ≥ 2000	16 (9.20%)	80 (34.48%)		
Total	174	232		
**Father’s Financial Support**			128.7	0.00
No	78 (44.83%)	0 (0)		
Yes	96 (55.17%)	232 (100%)		
Total	174	232		
**Parity**			111.75	0.00
<3	174 (100%)	123 (53.02%)		
≥3	0 (0)	109 (46.98%)		
Total	174	232		
**Baby’s Age in Weeks**			113.40	0.00
1–10 weeks	91 (52.30%)	86 (37.07%)		
11–24 weeks	83 (47.70%)	71 (26.80%)		
25–48 weeks	0 (0)	105 (45.26%)		
Total	174	232		
**Method of Delivery**			156.39	0.00
Normal vaginal delivery	83 (47.70%)	265 (100%)		
Caesarean section	91 (52.30%)	0 (0)		
Total	174	232		
**Planned Baby**			243.60	0.00
Unplanned baby	126 (72.41%)	0 (0)		
Planned baby	48 (27.59%	232 (100%)		
Total	174	232		
**Baby’s Health**			30.53	0.00
Healthy	174 (100%)	195 (84.05%)		
Sickly	0 (0)	37 (15.95%)		
Total	174	232		
**Baby’s Preferred Sex**			227.89	0.00
Girl	0 (0)	163 (70.26%)		
Boy	125 (71.84%)	67 (28.88%)		
No preferred sex	49 (28.16%)	0 (0)		
I didn’t want a baby	0 (0)	2 (0.86%)		
Total	174	232		
**Baby Breastfed**			124.67	0.00
No	76 (43.68%)	0 (0)		
Yes	98 (56.32%)	232 (100%)		
Total	174	232		
**Currently Have Partner**			95.71	0.00
No	61(35.06%)	0 (0)		
Yes	113 (64.94%)	265 (100%)		
Total	174	232		
**Support From Partner**			216.53	0.00
No	116 (66.67%)	0 (0)		
Yes	58 (33.33%)	232(100%)		
Total	174	232		
**Partner has Other Sexual Partners**			190.01	0.00
No	174 (100%)	76 (32.76%)		
Yes	0 (0)	156 (67.24%)		
Total	174	232		
**Has Support in Difficult Times**			47.89	0.00
No	33 (81.03%)	0 (0)		
Yes	141(81.03	232 (100%)		
Total	174	232		
**Who Supports**			224.40	0.00
No one	0 (0)	31 (10.0%)		
Family	15 (8.62%)	159 (68.53%)		
Husband/partner	115 (66.09%)	42 (18.10%)		
Friend	44 (25.29%)	0 (0)		
Total	174	232		
**Threatened to Hit**			0.26	0.61
No	154 (88.51%)	209 (90.09%)		
Yes	20 (11.49%)	23 (9.31%)		
Total	174	232		
**Partner Drinking Alcohol**			26.72	0.00
No	73 (41.95%)	43 (18.53%		
Yes	101 (58.05%)	189 (81.47%)		
Total	174	232		
**Has Experienced Severe Stress**			378.25	0.00
No	167 (95.98%)	0 (0)		
Yes	7 (4.02)	232 (100%)		
Total	174	232		
**Reason for Stress**			41.94	0.00
Severe financial crisis	91 (52.30%)	63 (96.92%)		
Death of a close person	26 (14.94%)	0 (0)		
Illness	22 (12.64%)	1(1.54%)		
Cheating and not supportive	22 (12.64%)	0 (0)		
Changed jobs and place	8 (4.60%)	0 (0)		
Regret, unhappiness, and life-threatening crime	5 (2.87%)	1 (1.54%)		
Total	174	232		

Chi2- chisquare test.

**Table 4 ijerph-17-06980-t004:** Logistic regression for the variables associated with postnatal depression.

Variable	Odds Ratio	Std. Error	Z	P > |Z|	95% Confidence Interval
Household income	1.44	0.11	4.57	0.00	1.23–1.67
Baby’s age in weeks	1.34	0.07	5.41	0.00	1.09–1.24
Currently have partner	0.45	0.11	−3.4	0.00	0.28–0.71
Support difficult times	76	76.5	4.30	0.00	10.57–546.51
Preferred sex	0.47	0.05	−6.82	0.00	0.37–0.58

P = level of significance; Z = z score.

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
