# Peer review of "The Need for Universal Screening for Postnatal Depression in South Africa: Confirmation from a Sub-District in Pretoria, South Africa"

_ijerph, 2020, doi:10.3390/ijerph17196980_

Round 1

Reviewer 1 Report

The purpose of this study was to determine the prevalence and predictive factors associated with postnatal depression in South Africa to establish the need to universal screening of postnatal depression in South Africa. This is a very important study and I am excited that it was conducted. However, there are a few major and minor points that need to be addressed.

Major points:
- While many studies use a cut-off score of ≥12 on the EPDS, the validation cut-off according to Cox et al (1987) is actually ≥13 for English-speaking women in the postpartum period (Cox J, Holden J, Sagovsky R. Detection of postnatal depression: development of the 10 item Edinburgh Postnatal Depression Scale. Br J Psychiatry 1987;150:782-6.). This error has been made repeatedly, but importantly a difference of just 1 point in the cut-off score has a significant non-trivial impact on the findings (Matthey S, Henshaw C, Elliott S, Barnett B. Variability in use of cut-off scores and formats on the Edinburgh Postnatal Depression Scale implications for clinical and research practice. Arch Women Ment Health 2006;9:309-15.) The reference the authors cite here is for the EPDS validation study is for the use of the EPDS during pregnancy. I suggest the authors redo the analysis with the validated cut-off of ≥13.
- A Chi-square test is really a descriptive test so while it may show the strength of a relationship, it is not appropriate to model the determinants of and predict the likelihood of an outcome. It does not take confounding factors and multiple testing into account. The authors should decide a priori, which predictors and confounder to include in the regression model and present that is the main analysis.

Below are a number of smaller points.

Abstract:
1. There was a statistical difference between women who were depressed and those that were not. – difference in what?
2. preferred gender? Of the mother or the baby? If the baby it should be sex
3. “the results continue the need for routine screening for postnatal depressive symptoms in primary health care facilities, which are used by the majority of women in South Africa.” – sentence incomplete
Introduction
1. 60% seems very high as a prevalence rate and the review included all studies irrespective of quality so may not be fully reliable.
2. Replace gender of baby with sex throughout – at this point it is still the sex of the baby because sex is based on biological attributes

Methods
3. Why was every second woman rather than every woman asked to participate?
4. How was the EPDS translated to ensure validity? Did you the authors use back-translation?

Results
5. What was the standard deviation of the EPDS scores?
6. This is semantics but a Chi-Square is used to test for differences between women with PND and women without PND. To test the association, you should run a regression model.
7. This feeds into one of the main points but how were the variables for the logistic regression selected? The result section suggests that it “run on those factors that were associated with PND on Chi Square” but table 4 only presents five of the 24 variables which differed significantly between the two groups. Related to that, what confounders were considered?
8. The regression results may be presented with 2 digits after the decimal point, no need for all those digits.

Author Response

Addressing reviewer 1

Reviewer 1

Section

Comment

How it was addressed

Major comments

- While many studies use a cut-off score of ≥12 on the EPDS, the validation cut-off according to Cox et al (1987) is actually ≥13 for English-speaking women in the postpartum period (Cox J, Holden J, Sagovsky R. Detection of postnatal depression: development of the 10 item Edinburgh Postnatal Depression Scale. Br J Psychiatry 1987;150:782-6.). This error has been made repeatedly, but importantly a difference of just 1 point in the cut-off score has a significant non-trivial impact on the findings (Matthey S, Henshaw C, Elliott S, Barnett B. Variability in use of cut-off scores and formats on the Edinburgh Postnatal Depression Scale implications for clinical and research practice. Arch Women Ment Health 2006;9:309-15.) The reference the authors cite here is for the EPDS validation study is for the use of the EPDS during pregnancy. I suggest the authors redo the analysis with the validated cut-off of ≥13.
- A Chi-square test is really a descriptive test so while it may show the strength of a relationship, it is not appropriate to model the determinants of and predict the likelihood of an outcome. It does not take confounding factors and multiple testing into account. The authors should decide a priori, which predictors and confounder to include in the regression model and present that is the main analysis.

Data was re-analysed using a cut-off score of 13 and the results section was re-written.

Abstract

1. There was a statistical difference between women who were depressed and those that were not. – difference in what?

Sentence removed

2. Preferred gender? Of the mother or the baby? If the baby it should be sex

Changed gender (of the baby) to sex throughout document

3. “the results continue the need for routine screening for postnatal depressive symptoms in primary health care facilities, which are used by the majority of women in South Africa.” – sentence incomplete

Sentence completed

Introduction

1. 60% seems very high as a prevalence rate and the review included all studies irrespective of quality so may not be fully reliable. 

Re-analysed the data using a cut-off of 13

2. Replace gender of baby with sex throughout – at this point it is still the sex of the baby because sex is based on biological attributes

Replaced gender with sex throughout document

Methods

3. Why was every second woman rather than every woman asked to participate?

This was for random selection of participants

4. How was the EPDS translated to ensure validity? Did you the authors use back-translation?

There is an ongoing bigger national study to screen for postnatal depression in the Department of Public Health and the translation and back-translation of the questionnaires into different indigenous languages were done previously. The student/2nd author selected the languages which were applicable to the community she was conducting the study on.

Results

5. What was the standard deviation of the EPDS scores?

Standard deviation inserted

6. This is semantics but a Chi-Square is used to test for differences between women with PND and women without PND. To test the association, you should run a regression model.

Regression model run and results presented and discussed

7. This feeds into one of the main points but how were the variables for the logistic regression selected? The result section suggests that it “run on those factors that were associated with PND on Chi Square” but table 4 only presents five of the 24 variables which differed significantly between the two groups. Related to that, what confounders were considered?

All the variables that were associated with PND on Chi square were included in the logistic regression  model, and only the  variables strongly associated with PND were extracted and included in table 4, hence only 5 of them

8. The regression results may be presented with 2 digits after the decimal point, no need for all those digits

Corrected and results presented with 2 digits after the decimal points

Reviewer 2 Report

This is, in summary, an interesting paper aimed to determine the prevalence of, as well as factors associated with postnatal depression in a sample of clinic attendees in a sub district in Tshwane, South Africa. The authors reported that the majority of the females had scores which are indicative of postnatal depressive symptoms. There was a difference between females who were depressed and those that were not. According to a multivariate analysis, postnatal depressive symptoms were significantly associated with who the mother lives with, preferred 22 gender, household income, 23 and baby’s age in weeks.

The authors may find as follows my main comments/suggestions.

First, when throughout the Introduction section, the authors correctly focused on depression which is associated with a significant disability and psychosocial impairment worldwide, they could even briefly describe the involvement of sensory perception whch is similarly implicated in emotional processes and clinical outcomes. Importantly, the unique sensory processing patterns in subjects at risk for suicide have been reported. Hyposensitivity or hypersensitivity may be "trait" markers of individuals with problem behaviors and interventions should refer to the individual unique sensory profiles and their behavioral and functional impact in the context of real life. Thus, given the above information, my suggestion is to include throughout the manuscript, the paper published in 2016 on Psychiatry Res (PMID: 26738981). In addition, subjects with a history of childhood maltreatment may be even at increased risk of suicidal behavior. In particular, the relation between moderate-severe childhood maltreatment, gender, depression, and suicidal behaviors has been demonstrated. Importantly, the exposure to abuse and neglect as a child may increase the risk to develop both symptoms of depression and higher suicidal risk. Thus, in order to briefly discuss this topic (although i understand that the link between depression, childhood maltreatment, and suicidal behavior is not the main topic of the present manuscript), i suggest to cite, within the main text, the paper published in 2017 on Frontiers in Psychiatry (PMID: 25169890).

In addition, the most relevant aims/objectives and hypotheses of the present study should be described in amore detailed manner within the main text.

Importantly, inclusion/exclusion criteria need to be reported extensively.

Moreover, the authors should immediately present and discuss, in the first lines of the Discussion section, their most relevant study findings. Conversely, they seem to focus with redundance on what existing studies conducted in South Africa documented about this topic that should have been stressed elsewhere.

Furthermore, the major shortcomings/limitations of this paper need to be better discussed as the description of the main caveats has been not included within the main text.

Finally, what is the take-home message of this study? While the authors suggested that the present results continue the need for routine screening for postnatal depressive symptoms in primary health care facilities, which are used by the majority of females in South Africa, they failed, in my opinion, to provide some conclusive remarks of their paper. Here, more details/information are needed.

Finally, the manuscript needs to be reviewed by a native English speaker for the quality of the language.

Author Response

Reviewer 2

First, when throughout the Introduction section, the authors correctly focused on depression which is associated with a significant disability and psychosocial impairment worldwide, they could even briefly describe the involvement of sensory perception which is similarly implicated in emotional processes and clinical outcomes. Importantly, the unique sensory processing patterns in subjects at risk for suicide have been reported. Hyposensitivity or hypersensitivity may be "trait" markers of individuals with problem behaviors and interventions should refer to the individual unique sensory profiles and their behavioral and functional impact in the context of real life. Thus, given the above information, my suggestion is to include throughout the manuscript, the paper published in 2016 on Psychiatry Res (PMID: 26738981). In addition, subjects with a history of childhood maltreatment may be even at increased risk of suicidal behavior. In particular, the relation between moderate-severe childhood maltreatment, gender, depression, and suicidal behaviors has been demonstrated. Importantly, the exposure to abuse and neglect as a child may increase the risk to develop both symptoms of depression and higher suicidal risk. Thus, in order to briefly discuss this topic (although i understand that the link between depression, childhood maltreatment, and suicidal behavior is not the main topic of the present manuscript), i suggest to cite, within the main text, the paper published in 2017 on Frontiers in Psychiatry (PMID: 25169890).

Involvement of sensory perception integrated into manuscript (see lines 45 to 50)

The publication by Serafini et al (2017) integrated, and the link between childhood maltreatment and suicidal behavior integrated (see lines 55 to 62)

In addition, the most relevant aims/objectives and hypotheses of the present study should be described in a more detailed manner within the main text.

Included, see lines 86 to 88

Importantly, inclusion/exclusion criteria need to be reported extensively.

Included, (see lines 110 to 115)

Moreover, the authors should immediately present and discuss, in the first lines of the Discussion section, their most relevant study findings. Conversely, they seem to focus with redundancy on what existing studies conducted in South Africa documented about this topic that should have been stressed elsewhere.

First lines of the discussion re-written to focus on the study findings (see lines 205 to 210 )

Furthermore, the major shortcomings/limitations of this paper need to be better discussed as the description of the main caveats has been not included within the main text.

Discussed, see lines 268 to 276

Finally, what is the take-home message of this study? While the authors suggested that the present results continue the need for routine screening for postnatal depressive symptoms in primary health care facilities, which are used by the majority of females in South Africa, they failed, in my opinion, to provide some conclusive remarks of their paper. Here, more details/information are needed.

Re-wrote the conclusion and recommendation sections (see lines 253 to 261)

Finally, the manuscript needs to be reviewed by a native English speaker for the quality of the language.

Submitted to MDPI language reviewers

Round 2

Reviewer 1 Report

Thank you for addressing my comments.

Reviewer 2 Report

In the revised manuscript, the authors addressed most of the major questions raised by Reviewers improving the main structure of the present manuscript. I have no further additional comments.